# Learning to Look by Self-Prediction

**Matthew Koichi Grimes     Joseph Modayil     Piotr Mirowski     Dushyant Rao     Raia Hadsell**
DeepMind, London UK
{mkg,modayil,piotrmirowski,dushyantr,raia}@deepmind.com

## Abstract

We present a method for learning active vision skills, for moving the camera to observe a robot's sensors from informative points of view, without external rewards or labels. We do this by jointly training a visual predictor network, which predicts future returns of the sensors using pixels, and a camera control agent, which we reward using the negative error of the predictor. The agent thus moves the camera to points of view that are most predictive for a target sensor, which we select using a conditioning input to the agent. We show that despite this noisy learned reward function, the learned policies are competent, avoid occlusions, and precisely frame the sensor to a specific location in the view, which we call an emergent fovea. We find that replacing the conventional camera with a foveal camera further increases the policies' precision.

## 1   Introduction

Human vision does not closely resemble computer vision, as commonly practiced in embodied RL and robotics. Human vision has moving eyes, fovea, movements such as saccades to frame targets in view, and smooth pursuit to track them [8]. By contrast, vision can be a source of training instability in RL [3], leading robotics researchers to avoid it altogether and use object features or untrained vision modules instead [25], or to stabilize the camera by fixing it to the environment, limiting the agent to stay within the camera's field of view [19].

An agent that has learned to visually frame objects in a consistent image location actively reduces the image-space variance attributable to object position. Locking down the object's position within the image could simplify the acquisition of visually-guided manipulation policies, as they can then focus on the manipulation aspect of the policy. This intuition has recently seen evidence in robotic manipulation research on hand-mounted cameras, where the object appears in a consistent position as the hand is about to grasp it [14, 4, 11, 35, 15].

By contrast, humans benefit from decoupling the kinematic chain of the eye from those of the limbs, allowing them to flexibly choose their visual focus in highly dynamic tasks ranging from fielding baseballs [21] to traversing challenging terrain [20]. These visual policies all share common building blocks in the form of fixation and tracking (saccades and smooth pursuit). One inspiration to our paper is the question: if fixation is a general visual skill that is key to acquiring more task-specific visual policies, how can an agent learn it independently of specific tasks? This paper demonstrates how an embodied agent may acquire visual skills in the absence of external rewards.

Previous work has investigated related yet distinct ideas. Yang et al. [39] propose embodied perception where an agent learns to move around an object to perceive it better. Nilsson et al. [24] study embodied vision in a navigating agent solving a semantic segmentation task. Gärtner et al. [10] use RL to select the best viewpoint in a panoptic camera rig for human pose estimation. Pirinen et al. [28] train an agent to seek points of view for 3D reconstruction. Jayaraman & Grauman [16] use active vision to

4th Workshop on Shared Visual Representations in Human and Machine Visual Intelligence (SVRHM) at the Neural Information Processing Systems (NeurIPS) conference 2022. New Orleans.

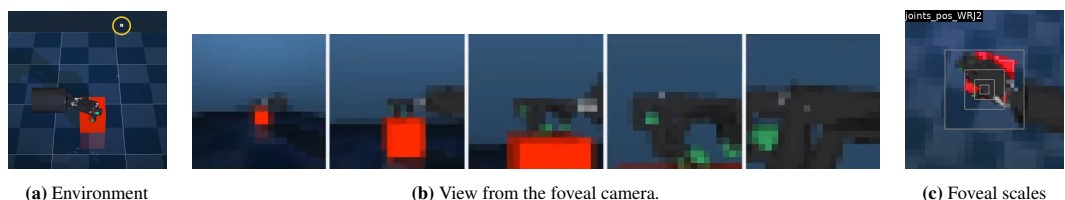

| (a) Environment | (b) View from the foveal camera. | (c) Foveal scales |

**Figure 1:** The environment, with hand, box prop, and foveal camera. Fig. 1b shows the foveal image, a stack of 5 RGB images, with shape (5, height, width, 3). The images have the same dimension, but cover differing fields of view: 90, 45, 22, 11, and 5 degrees. Fig. 1c shows another foveal view.

forecast the effects of a motion. More generally, previous works [30, 17] also learn to look around to efficiently gather information about the agent's surroundings.

## 2 Methods

Our environment (fig. 1a) uses the open-source (Apache 2.0) MuJoCo [37] physics simulator, in which we place a camera at the end of an invisible armature (the *camera bot*), controlled by the camera agent. This shares the scene with a manipulator, and a block prop randomly positioned within reach of the fingers, which gives the fingertip touch sensors something to touch. Both the camera bot and manipulator are driven by velocity control, i.e. proportional-integral-derivative (PID) control in which the actions specify a target velocity for each actuator.

We treat the camera bot and manipulator as belonging to one robot that is conceptually split into two separate entities. The manipulator runs a fixed random behavioral policy throughout the experiment. Sensors on the manipulator are available during training, to train the predictor. The predictor serves as an endogenous reward function for simultaneously training the agent, which controls the camera bot. The training process only modifies the behavior of the camera bot, not the manipulator.

The manipulator is a model of the 20-DOF tendon-driven hand by Shadow Robotics [32, 29]. We randomly drive the manipulator using Perlin noise [27]. The camera bot has four DOFs: `azimuth`, `elevation`, `distance`, and `yaw`. The first three position the camera in spherical coordinates around the manipulator, and the last DOF rotates it in place. All DOFs are initialized randomly, which causes the hand to be out of view half the time.

**Cameras:** The camera bot can be equipped with a conventional camera, or a foveal one [5, 12, 6, 7]. Like Mnih et al. [22], we implement the foveal camera as $N$ superimposed conventional cameras with the same image dimensions, differing only in their field of view. Fig. 1b shows an example of the resulting images as seen by the agent, and fig. 1c visualizes their relative fields of view by stacking them on top of each other. In our experiments, the foveal camera has $N = 5$ cameras with $21 \times 21$ pixels each. Camera 1 has a FOV of 90 degrees in the vertical and horizontal directions, camera 2 has a FOV of 45 degrees, and so on, down to camera 5 with a FOV of 5.26 degrees ($90/2^4$). The conventional camera is $21 \times 21$ pixels, with a FOV of $90 \times 90$ degrees.

**Predictor network:** The predictor network shares no weights with the policy or critic networks in the agent. It is a convolutional residual network (resnet) followed by a multi-layer perceptron (MLP). Its architecture and layer sizes are taken from IMPALA's [9] convolutional residual network, replacing the LSTM at the end with an MLP with hidden layer sizes $(512, 512, 256)$. This predictor network takes two consecutive images as input, and outputs predictions for all potential target sensors. Like the critic network, its predictions take the form of discrete distributions of estimated returns, of sensor values rather than rewards. This was inspired by multi-timescale nexting [23], and the predictor network can be thought of as a multi-headed GVF [34] with distributional outputs. The errors from these predictions can be thought of as a form of surprise, and driving behavior to maximize prediction errors has been suggested as a form of curiosity [38]. In a sense, our agent does the opposite of this, as it is rewarded for minimizing error. Instead of exploring by maximising short-term surprise, it explores the state space through the diversity of its sensor targets.

**Agent networks:** The agent consists of a policy network and critic network. Like the predictor network, each of these is an MLP stacked on an IMPALA-style convolutional resnet. The critic and policy networks share a resnet, but have different MLP heads, with hidden layer sizes $(256, 256, 256)$

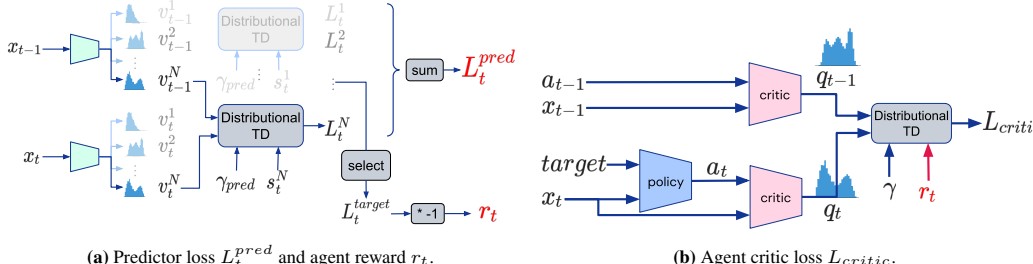

**(a)** Predictor loss $L_t^{pred}$ and agent reward $r_t$.

**(b)** Agent critic loss $L_{critic}$.

**Figure 2:** a) Distributional TD losses $L_t^i$ are computed for each sensor $s_t^i$, in a loop over $N$ sensors (the figure shows only the $N$-th sensor). The predictor loss $L_t^{pred}$ is the sum of the $N$ distributional TD losses. A different *target* sensor is chosen at the start of each episode, whose prediction error $L_t^{target}$ is used as a negative agent reward $r_t$. b) The D4PG [1] critic loss $L_{critic}$ depends on the learned agent reward $r_t$. The *target* sensor (see Section 2.1) is specified as a one-hot vector, fed to the policy as a conditioning input.

and $(512, 512, 256)$, respectively. Both networks take a mix of images and other inputs. The images are fed through the resnet, and the other inputs are concatenated with the result and fed to the MLP. The policy network takes as input the observation (two consecutive image frames), and a one-hot vector specifying the target. It outputs bounded continuous target velocities for the camera bot's four joints (section 2). The state-action critic additionally takes the action, and outputs a discrete distribution of the estimated return. The agent's objective is to minimize the predictor network's error for the target sensor, by modifying the policy network to move the camera to look at the associated body part.

## 2.1 Training

For each batch of transitions sampled from the replay buffer, we compute losses for the predictor, agent critic, and agent policy networks, then perform gradient updates on all three.

To train the predictor, we compute the prediction error of sensor $i$ using the distributional TD loss [2] from the distributed policy gradient method [1]. This is analogous to the standard TD error ($\delta = r_t + \gamma v_t - v_{t-1}(x)$), except that values $v$ are represented not by scalars, but by discrete distributions over the range of possible return values. Instead of computing Q-values of reward, we estimate the expected return, $v^i$, of sensor reading $s_t^i$:

$$L_t^i = DistributionalTD(s_t^i, \gamma_{pred}, v_t^i, v_{t-1}^i). \tag{1}$$

The predictor discount $\gamma_{pred}$ is separate from the discount $\gamma$ used for training the critic. Setting $\gamma_{pred} = 0$ amounts to performing next-frame prediction, while setting it to larger values predicts its future sum over a decaying time window with half-life $h = \Delta t \frac{ln(0.5)}{ln(\gamma_{pred})}$. It is possible to predict over multiple time windows as in Horde [34], which may be useful in environments with predictable dynamics over multiple frames. For our environment, where the camera and manipulator have little momentum, we use decay $\gamma_{pred}$ chosen to have a short half-life of 0.1s. The predictor network's loss is then the sum of prediction losses across all target sensors, $L_t^{pred} = \sum_i L_t^i$.

The camera agent networks share no parameters with the predictor. Each episode randomly chooses a proprioceptive sensor on the manipulator, to serve as the camera agent's *target* for that episode. The camera agent's task is to position the camera in a manner that reduces the predictor's error for that target sensor. We therefore define the reward to be $r_t = -L_t^{target}$, where $L_t^{target}$ is the prediction error for that episode's target sensor, on timestep $t$. This is a dense reward, as it is available on every timestep, but a noisy one, as it is learned. The agent critic loss is analogous to eq. 1, substituting $r_t$ for sensor reading $s_t^i$, $\gamma$ for $\gamma_{pred}$, and critic outputs $q_t$ and $q_{t-1}$ for predicted sensor returns $v_t$ and $v_{t-1}$. We train the policy network using the deterministic policy gradient loss [33].

Training takes 3 days to converge. We generate data by running 512 instances of the simulator across as many machines, which feed training data to a single learner machine powered by a TPU.

# 3 Experimental Results

We present results from our trained agent, evaluated without the exploration noise used during $\epsilon$-greedy training.

## 3.1 The trained camera policy improves target prediction accuracy

| | Little finger touch | Forefinger root joint angle | Thumb root joint angle | Wrist flexion angle | Wrist deviation angle |
|---|---|---|---|---|---|
| Blind (c) | $0.680 \pm 0.020$ | $4.12 \pm 0.065$ | $4.11 \pm 0.073$ | $3.84 \pm 0.059$ | $4.08 \pm 0.051$ |
| Random (c) | $0.667 \pm 0.067$ | $4.16 \pm 0.034$ | $4.16 \pm 0.056$ | $3.51 \pm 0.069$ | $3.61 \pm 0.050$ |
| Random (f) | $0.648 \pm 0.048$ | $4.07 \pm 0.033$ | $4.09 \pm 0.068$ | $3.41 \pm 0.054$ | $3.58 \pm 0.077$ |
| Ours (c) | $0.582 \pm 0.062$ | $3.24 \pm 0.070$ | $2.98 \pm 0.061$ | $1.73 \pm 0.054$ | $2.17 \pm 0.048$ |
| Ours (f) | $0.606 \pm 0.046$ | $3.11 \pm 0.050$ | $2.75 \pm 0.071$ | $1.65 \pm 0.047$ | $1.93 \pm 0.051$ |
| Oracle (c) | $0.480 \pm 0.020$ | $2.80 \pm 0.096$ | $2.53 \pm 0.025$ | $1.45 \pm 0.017$ | $1.62 \pm 0.026$ |
| Oracle (f) | $0.587 \pm 0.022$ | $2.80 \pm 0.030$ | $2.53 \pm 0.032$ | $1.45 \pm 0.009$ | $1.66 \pm 0.010$ |

**Table 1: Target sensor's prediction error at episode end (lower is better).** The "(c)" and "(f)" indicate conventional or foveal camera. *Blind* and *Oracle* give upper and lower bounds to the error, and *Random* shows the prediction error of a randomly posed camera. Error is measured as the TD error for predictions given as distributions over the range of possible return values. Confidence bounds indicate $\pm 1.96\sigma$, calculated from 10 runs with different RNG seeds.

Table 1 shows the target's prediction error at the end of the episode (lower is better). It compares trained agents against the following baselines:

**Blind:** The *Blind* baseline is a predictor trained on a camera pointed away from the hand. It can do no better than learn to output each sensor's prior distribution, and serves as an upper bound to the expected agent prediction error.

**Oracle**: For a lower bound on the prediction error, we run a sweep over a series of hand-chosen fixed camera poses surrounding and looking at the hand, training a separate predictor for each pose. The *Oracle* entries show the minimum prediction error over all viewpoints for that target sensor. The Oracle benefits from not only specializing to a single point of view, but also from not moving, which significantly reduces data variance and improves prediction error even in the moving-camera agent. In practice, moving agents cannot always maintain a static view, nor can an agent with multiple static cameras usually know a priori which camera will yield the most accurate predictions.

**Random**: The *Random* agent is our agent with its policy and critic learning rates set to zero. The camera spawns randomly, as usual, but hardly moves thereafter (brownian actions do little to move the camera bot, which has high inertia). The predictor must learn to predict from the resulting random camera views. These views are mostly static, advantageously reducing input variance in a similar manner to the still images of the Oracle. Our agent outperforms the Blind and Random baselines by a statistically significant margin.

## 3.2 The trained camera policy frames the subject

A consistent outcome in our experiments is that camera agents learn to to place the target in a particular position in the screen. This position can depend on the target, and varies from one training run to another, though it is usually near the center (fig. 3a). This is not an instance of the camera agent having memorized a particular set of preferred values for its own joint angles. The camera agent is unaware of the camera bot's joint angles, as the only inputs are pixels and the one-hot vector specifying the target sensor. Furthermore, it controls the camera bot joints by velocity control, whereas a position-controlled camera may learn to output a constant target camera pose without regard to input. Figure 3b shows the emergence of this *behavioral fovea* in a conventional (non-foveal) camera. Figure 3c show that agents with foveal cameras learn to frame the subject more precisely, in the center.

## 3.3 The camera policy adopts distinct camera positions for different sensor targets

Section 3.2 showed that the policy learns to orient the camera to frame the target sensor at a specific image location. While it is obviously important to look in the right direction, looking from the right

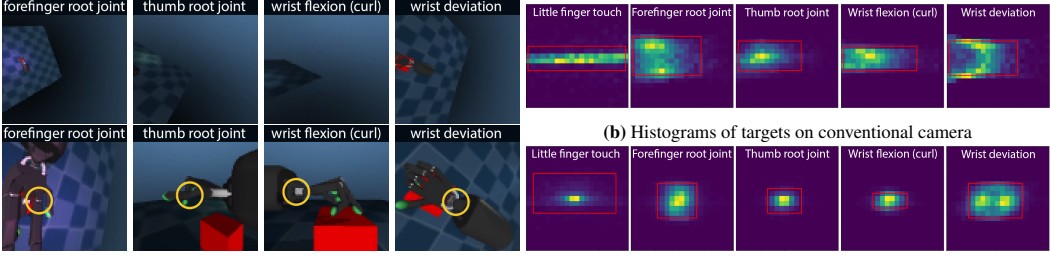

**(a)** First (top row) and last (bottom row) frames

**(b)** Histograms of targets on conventional camera

**(c)** Histograms of targets on foveal camera

**Figure 3:** Fig. **a**: the first and last frames (top and bottom row) from episodes where the sensor in the caption is the visual target. The figure highlights the target sensor with yellow circles (the agent does not see this). Note that the agent chooses to observe the wrist flexion angle (3rd column) from the side, from where it is most visible, while it observes the wrist's side-to-side deviation angle (4th column) from above. Fig. **b** and **c** show histograms of target location at the end of the episode, in image space, for conventional and foveal agents. The red box covers 2.5% to 97.5% of the cumulative probability distribution along each axis. The histograms accumulate the final target position over episodes collected after convergence, or the last 8 days out of an 11 day training run. This amounts to a total of 25000 episodes, or roughly 5000 episodes per target.

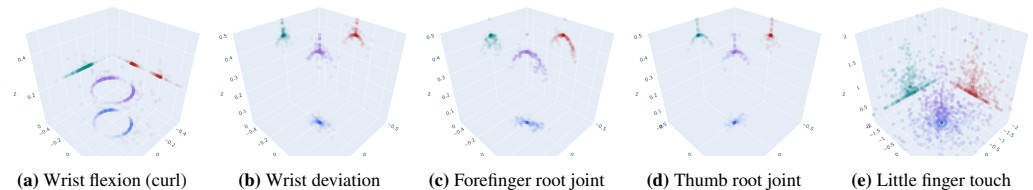

**(a)** Wrist flexion (curl)  **(b)** Wrist deviation  **(c)** Forefinger root joint  **(d)** Thumb root joint  **(e)** Little finger touch

**Figure 4:** Positions of the camera at the end of the episode, for different target sensors. Positions are in purple, with their projections to axis-aligned planes shown in red, green, and blue.

position also matters. Fig. 4 shows the distribution of the camera position at the end of the episode for a single trained agent, showing a separate plot for each sensor target. Figure 4a shows that the camera has learned to observe the wrist flexion in profile, i.e., along the axis of rotation, from which the visual flow of flexing the wrist is most apparent. It observes the wrist from one side or the other, hence the bimodal distribution seen in the red projection. By contrast, observing the same wrist joint, but predicting its side-to-side deviation angle, behooves the agent to adopt a top-down view, as shown in fig. 4b. The differences in distribution for the camera positions when observing the finger root joints (fig. 4c, fig. 4d are more subtle, observable in the blue projection along the floor. Fig. 4e shows the agent to be more position-agnostic when targeting the little finger's touch sensor, due to its noisier predictions. We discuss the reasons for this in section 5.

## 3.4 The camera policy learns to circumnavigate occlusions

We introduced a randomized occluder to our training environment to test the generality of our self-supervised training. The occluder is a flat rectangle with random color, position, and dimensions. We uniformly sampled its height and width from the range $[0.5, 1.0]$, and sampled its position in spherical coordinates from $azimuth \sim [-\pi, \pi]$, $elevation \sim [0, \pi]$, $distance \sim [.4, .6]$. These spherical coordinates are roughly centered around the hand. For reference, the size of the environment's floor is $2 \times 2$. To maximize its effectiveness as an occluder, we orient the board to face the hand. Fig. 5 shows two trajectories of the camera agent, starting from the same initial conditions. The resulting camera policy is able to sidestep the occluder when it blocks the camera's view, thereafter exhibiting similar viewpoint preferences as in the unoccluded case.

## 4 Discussion

Many animals do not have a fovea. Some, like rats, have a roughly even resolution across a nearly spherical field of view. When all directions are seen with equal acuity, one might ask if rats need to

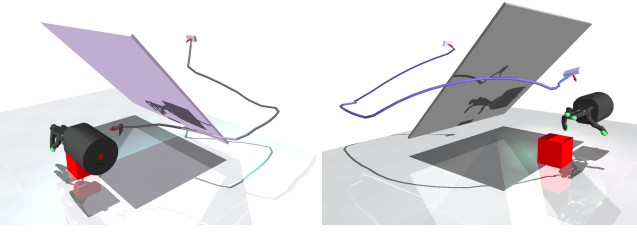

**(a)** Wrist flexion is viewed in profile.  **(b)** Wrist deviation is viewed from above.

**Figure 5:** Occlusion-avoidant trajectories, by the camera agent trained with random occlusions.

move their eyes to look *at* subjects at all. Yet they do, centering and fixating the subject in a specific area of the retina [13]. Like the rat, we find that our non-foveal agents exhibit framing behavior, positioning the subject in a specific area of the image, giving rise to a "behavioral fovea" even in the absence of a physical one.

This suggests that the learning dynamics encourage a positive feedback loop between the predictor and agent: the predictor improves its expertise in a specific region of the field of vision, and the agent learns to move the relevant subject into this visual region to receive the better prediction reward, and thus provides even more training data to further improve the predictor. This behavior is in contrast to the usual emphasis placed on the data-hungry problem of learning position-invariant or position-equivariant representations, a practice originating in computer vision research with datasets of still images, rather than with an active camera [18]. That said, even static facial recognition has been shown to benefit from normalizing the facial feature locations [36], a form of framing.

**Future work** Giving our agent a foveal camera incentivizes it to position the subject more precisely at the center of the image. One direction for future work is to investigate whether the agent can maintain this centering as the target body part makes large movements, resulting in tracking behavior from the camera as a side-effect of framing. In this work, we limit the target sensors to a single hand, but a realistic body provides a rich variety of visual scales and distances, from shoulders to toetips. We posit that these present a means to learn a rich repertoire of visual fixation and tracking skills without the need to design their reward functions. These skills could serve as a basis for exploration while learning higher-level skills, as demonstrated with SAC-X [31], which used manually designed reward functions. Parisi et al. [26] show that visual RL agents benefit from pretraining on static image datasets. Self-supervised RL presents a means of feature learning that is, in a lifelong learning sense, more natural than using human-annotated image datasets.

**Limitations** For N targets, the time spent training to target any one of them is 1/N of the total training time, leading to poor scaling of the needed total training time as N increases. We plan to investigate target relabeling to mitigate this. The manipulator's random policy leads to infrequent touch sensor activations, leading to a label imbalance for the touch predictor that makes targeting touch sensors more difficult to learn than with other sensor types. We plan to address this label imbalance by learning a policy for the hand that maximizes the target sensor's entropy.

## 5   Conclusion

In this paper we demonstrate a self-supervised means of learning to fixate a camera at areas that are predictive to a quantity of interest, such as a sensor reading. We do so in the context of a simulated robot, learning to fixate on various parts of its own body by predicting its own sensors. The learned policies are performant relative to baselines, precise, and able to circumnavigate occlusions. The precision increases when the agent is provided with a foveal camera.

## Acknowledgments and Disclosure of Funding

Funding by DeepMind. Special thanks to Bobak Shahriari, Matt Hoffman, and Piotr Trochim for technical help, Adam Kosiorek, Patrick Pilarski, and Nicolas Heess for internal review, and Joel Leibo and Kimberly Stachenfeld for neuroscience consultation.

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
