# OpenReview forum: "Learning to Look by Self-Prediction"
_NeurIPS.cc/2022/Workshop/SVRHM — SVRHM Oral_

### Official Review · Reviewer_ySJb · 2022-10-13
**A first step toward robotic eye-hand coordination learning**

**Rating:** 7
**Confidence:** 2

**Review:**

This paper describes a camera-control RL algorithm that seeks to position a camera so that the prediction error of a neural network predicting future somatosensory sensor readings from pairs of consecutive video frames would be minimized. Specifically, the somatosensory sensor readings are sampled from a robotic hand operated independently of the camera. The results indicate that the camera controller learns to place the target somatosensory sensor in a relatively fixed image position; The visual NN that processes the video signal learns to predict the target sensors significantly better than chance. Introducing a foveated camera promotes the positioning of the target sensor at the fovea.

This report is well-written, the experimental evaluation is convincing, and the model and associated methods are described in sufficient detail (except for the training parameters, which should be included in the camera-ready version).

As a potential biological model of eye-body coordination, the setup is limited since predicting somatosensory sensor readings is not a sufficient objective for guiding gaze during manual manipulation. In a more realistic setting, where the hand is to be guided to manipulate objects that may move (e.g., catching a ball), gaze control must serve objectives that relate to the external world rather than just proximal predictions. Furthermore, in humans, the control of the hands is not independent of the visual input.

Therefore, for future work, it might be interesting if the authors could build upon their methodology to extend the training of camera controllers to maximize task objectives and to incorporate visually-guided motor control of robotic hands.

I'm rating this submission as "accept". However, I rate it with low confidence since RL is not my field of specialty.

---

### Official Review · Reviewer_QkBa · 2022-10-14
**A very interesting paper**

**Rating:** 7
**Confidence:** 3

**Review:**

The paper asks an important question: how an agent can learn a policy of fixation and tracking without external feedback. It solves this by imposing an objective of reducing the estimation of states of an agent's own body parts (joints), which is naturally available to animals. Even if the agent randomly places its sensors, the policy network and a critic work together to learn a policy to move the camera and eventually place the joint of interest in view so that the prediction network can accurately infer the sensor values (joint angles).
The paper also found that when using foveal camera, the learned distribution of final placement of sensors in the camera is more concentrated than classical camera.

The work is important and interesting because it only uses the internal information available to the agent (i.e., the sensor values), which is close to the constraints faced by the brain.


Although it is a workshop paper, I think there are a few interesting questions to explore and a gap from learning a general active vision skill (2):
(1) Since the sensors are randomly placed, with a good chance they will not be in view at all at the beginning. It will be interesting to analyze the learned initial trajectory. For example, does the policy network learns to make initial big movement to explore areas previously unseen, until it sees any part of the sensors, and then gradually move towards it? Does it learn the relative location of different sensors so that it can move straight to the target sensor after it sees another sensor, based on their spatial relationship and the pose of the viewed sensor?
(2) I think there is still a big gap between the learned policy here and fixation and tracking in general, because the agent still needs to somehow generalize this policy to external objects, for which it does not have ground truth information of their pose/location, unlike its own sensor. This will at least prevent the calculation of loss for the prediction network, and the reward for the policy and critic networks. It would be nice to postulate the potential route towards that goal.

---

### Official Review · Reviewer_rLsc · 2022-10-14
**A revealing foray into the emergence of active sampling in a robotic system predicting its own state via differing modalities.**

**Rating:** 9
**Confidence:** 4

**Review:**

The paper details a virtual robotic system that learns, with reinforcement learning (RL) without an external reward, to move its camera onto a target sensor (controlling its arm) to predict the sensor's state (provided by the sensor). Impressively, the camera is moved into the appropriate position to read the sensor's state. For e.g. it would position the camera orthogonal to the plane containing the wrist's side-to-side deviation.

The methods are quite hard to understand for someone like me who does not have a background in RL. It would really help to have a schematic of what sensors are the targets and what exactly are the inputs and outputs of the predictor and actor-critic systems.

The foveation mechanism is unclear to me. Are all the five cameras, capturing different scales, provided as inputs to the predictor network? As opposed to just one camera in the conventional setting?

In the Predictor network section of the Methods, it is mentioned that the inputs are two consecutive images. It is unclear what function this serves. In contrast, in Fig. 2, it seems given the input from the camera at a timepoint, a prediction is made.

The controls mentioned in Table 1 are exhaustive and help qualify the behaviour of the system. This is much appreciated!

The observation that the camera is moved to a position best for inferring the state of a target sensor is cool. It would be useful to quantify this behaviour. For example, you could asses if the camera position is indeed orthogonal (or near-orthogonal) to the plane of deviation given a sensor.

The observation that the target sensor is placed near the fovea even with a conventional camera is interesting. The predictor network is a convolutional neural network. Why then would the camera need to almost fixate on the sensor? Do the predictions indeed get worse if you move the sensor further away from the centre of view of the camera? What part of the systems might be enforcing such a behaviour?

On the contrast with the "data-hungry" CV systems: I think this is a bit unfair as in that case the dataset does not allow for active sampling to the same extent as possible in your system. An assertion that this is why we need to move to simulated or natural environments where active sampling is possible would be relevant instead.

What I find super interesting is because the control network can move the camera into a position that gets a good viewpoint for state estimation, the prediction network could actually be smaller than what it might need to be if it has to predict the state from all viewpoints. This might *increase* the apparent capacity of the prediction network in terms of learning state estimations for more sensors, etc. The connection to two-stream systems in the brain might also be present. In visual processing, the dorsal stream is supposed to be in charge of spatial attention and eye movements and the ventral stream in charge of object recognition. Maybe the dorsal-ventral distinction maps quite well onto you control-prediction distinction. Just a thought :)

---

### Official Review · Reviewer_FRPZ · 2022-10-14
**An interesting model for focusing the center of vision on relevant body parts by interoceptive prediction.**

**Rating:** 8
**Confidence:** 3

**Review:**

This paper proposes a simple self-supervised learning model for focusing the center of vision on specific body parts of a robot in simulation. The problem is posed well, and the proposed solution seems to work satisfactorily. While the problem studied is a very interesting one, a few limitations prevent me from providing a higher score.

The agent for focusing the visual field is considered independent of the robotic arm itself which does not move very much during an episode, which leaves open the behavior under large self-movements as recognized in the manuscript. Furthermore, the convergence of the model seems quite slow to train despite this additional simplification.

It is also not clear to me why the convolutional layers are not shared between the predictor and the agent. This seems to go against the biologically-inspired design philosophy, and might also contribute to slower learning. A clarification on this point would be helpful.

The paper’s clarity could be improved by providing further details about the implementation of the training loss functions, perhaps in an appendix. A few explicit equations laying these out would help the reader very much. Some more details on the performance metrics would also improve clarity.

Pros:
1. Interesting problem to study, and a practical generalized value function-based architecture that works.

Cons:
1. Missing some implementational details.

---

### Official Review · Reviewer_DD4F · 2022-10-14

**Rating:** 8
**Confidence:** 4

**Review:**

1. Summary:
The authors make a case for using active cameras and show the emergence of a fovea-like response profile when attending a region of the scene. They use a hand simulator which is driven with random controls and a camera is placed on a manipulator. Their model comprises a predictor network that predicts the different sensor locations on the hand, meanwhile, a policy critic model which is used to control the arm minimizes this prediction error. Their results show the existence of localization of camera response that they expected for both the foveated and non-foveated cameras.

2. Quality and clarity:
The text is written clearly for the most part and the authors did a good job explaining the setup and the model. The paper is well-motivated, and the figures aid the understanding of the model. My one nitpick is that the task should be mentioned clearly. Perhaps with its own subsection or paragraph, that would have made it more amenable to read. Also, a_t was never mentioned in the text besides from figure 2, I am assuming that is the action for the critic.

3. Originality:
As far as I have read about the subject, the work is original and presents initial evidence for the utility of active cameras in scene understanding and shows fovea-like response localization.

4. Significance & Fit for the workshop:
The utility of an active camera can be understood as analogous to the saccadic eye movements and oculomotor responses for stabilization of the head when moving. Understanding the effect of these mechanisms on learning is important for the field, especially for embodied agents. With these pervasive connections, I think this paper's topic is very relevant to the workshop's agenda.

The authors clearly show the "emergent fovea", i.e. their model localizes the target in pixel space and confines the response which makes the visual response of the predictor and movement response of the agent work in tandem. I don't yet agree with the claim that active cameras counter position-equivariant approaches on grounds of the amount of required data. However, these ideas are worth digging into and perhaps more evidence can substantiate that claim.

The authors have satisfyingly demonstrated that a foveated camera works better in the context of an active camera. They also showed that an active camera setup where the policy-critic networks learn to reduce the prediction error works much better than a random policy giving a good proof of concept. For these reasons, I would recommend accepting the paper to the workshop.